# Old and New Facts and Speculations on the Role of the B Cell Receptor in the Origin of Chronic Lymphocytic Leukemia

**DOI:** 10.3390/ijms232214249

**Published:** 2022-11-17

**Authors:** Davide Bagnara, Andrea Nicola Mazzarello, Fabio Ghiotto, Monica Colombo, Giovanna Cutrona, Franco Fais, Manlio Ferrarini

**Affiliations:** 1Department of Experimental Medicine, University of Genoa, 16132 Genoa, Italy; 2IRCCS Ospedale Policlinico San Martino, 16132 Genoa, Italy

**Keywords:** immunoglobulin, chronic lymphocytic leukemia, ontogeny

## Abstract

The engagement of the B cell receptor (BcR) on the surface of leukemic cells represents a key event in chronic lymphocytic leukemia (CLL) since it can lead to the maintenance and expansion of the neoplastic clone. This notion was initially suggested by observations of the CLL BcR repertoire and of correlations existing between certain BcR features and the clinical outcomes of single patients. Based on these observations, tyrosine kinase inhibitors (TKIs), which block BcR signaling, have been introduced in therapy with the aim of inhibiting CLL cell clonal expansion and of controlling the disease. Indeed, the impressive results obtained with these compounds provided further proof of the role of BcR in CLL. In this article, the key steps that led to the determination of the role of BcR are reviewed, including the features of the CLL cell repertoire and the fine mechanisms causing BcR engagement and cell signaling. Furthermore, we discuss the biological effects of the engagement, which can lead to cell survival/proliferation or apoptosis depending on certain intrinsic cell characteristics and on signals that the micro-environment can deliver to the leukemic cells. In addition, consideration is given to alternative mechanisms promoting cell proliferation in the absence of BcR signaling, which can explain in part the incomplete effectiveness of TKI therapies. The role of the BcR in determining clonal evolution and disease progression is also described. Finally, we discuss possible models to explain the selection of a special BcR set during leukemogenesis. The BcR may deliver activation signals to the cells, which lead to their uncontrolled growth, with the possible collaboration of other still-undefined events which are capable of deregulating the normal physiological response of B cells to BcR-delivered stimuli.

## 1. Introduction

Chronic lymphocytic leukemia (CLL) is characterized by the clonal expansion of CD5^+^ and CD23^+^ B lymphocytes in the circulation, peripheral lymphoid organs and bone marrow [1,2,3]. CLL has a variable clinical course; a relative minority of cases progress rapidly to the advanced stages and often have unfavorable outcomes despite the recently developed therapies. In most patients, the disease remains indolent and sometimes does not require therapy. This dichotomy of courses, indicated by clinical observations, was not predictable until the advent of cellular and molecular studies [4,5]. Now, we know that the two patient groups differ in the expression of activation markers on the leukemic cells and in the presence/absence of cytogenetic alterations. More striking, perhaps, is the observation that patients with greater than 2% somatic mutations (mutated or M-CLL) in the genes encoding for the immunoglobulin heavy chain variable region (IGHV genes) have a more indolent clinical course and a more favorable outcome than those with less than 2% mutations of the same gene segments (unmutated or U-CLL), implying the involvement of these genes in the mechanisms of clonal expansion [6,7].

In the past, CLL was described as an accumulation of immune-incompetent small lymphocytes which proliferated slowly and had a low clearance. These features appeared to be consistent with the indolent nature of the disease and with its supposedly long cell survival in vivo [8]. Moreover, the concept of immune incompetence provided an explanation for the documented patient susceptibility to infections, given that an accumulation of cells, which are destined to protect the host from infections, would not serve this primary purpose if they were immune-incompetent. Progressive changes in this definition were brought about by several observations in vitro, including the capacity of CLL cells to proliferate, if stimulated appropriately, and undergo apoptosis when deprived of the adequate micro-environmental support. However, the major changes in the definition of CLL were brought about by kinetics studies, in which CLL cells were labeled with deuterated water and their survival in vivo could be measured [9,10,11]. These studies clearly demonstrated that leukemic cells could proliferate (thus contributing to clonal expansion) and compensate for the cell loss consequent to the death of a proportion of the clone, focusing the attention of researchers on the mechanisms that promoted leukemic cell proliferation. Differences in the in vivo kinetics of CLL clones were also demonstrated, with U-CLL clones proliferating more actively in vivo than M-CLL clones. This finding is in line with the observation that U-CLL cells have shorter telomeres and higher telomerase activity, again indicating more active turnover [12,13]. In addition, the finding that the CLL cells from a given patient were clonal, i.e., derived from the same progenitor cell as demonstrated by the utilization of a unique IGHV-IGHD-IGHJ gene rearrangement in each leukemic clone [14], provided a more refined explanation of CLL cell immune incompetence, given that the B cell Receptor (BcR) of the leukemic clone has a restricted specificity and is unable to cover the large repertoire required for the immune defense, even when assuming the normal functioning of the cells.

Frequently, an accelerated kinetics and/or prolonged cell survival in hematologic malignancies correlates with a common cytogenetic abnormality, characterizing the malignant cells. In Burkitt’s lymphoma (BL), in which the cells have a very high proliferation rate, there is a translocation of the *c-myc* oncogene from chromosome 8 to one of the chromosomes where the IG gene loci are located, i.e., to chromosome 14, 2 or 22 [15,16]. This translocation results in the juxtaposition of the *c-myc* gene to one of the IG gene loci, causing an upregulation of *c-myc* itself, since this oncogene is now transcribed at the same rate as that of the IG chains genes. *c-myc* encodes for a transcription factor which drives the cells into the cell cycle and, consequently, the abundant availability of MYC protein explains the very high proliferation rate of BL cells. Other situations are characterized by alterations of the apoptotic properties, as is the case for follicular lymphoma (FL), in which the *bcl-2* gene is translocated near the IG H chain gene locus, causing its upregulation. Since the BCL-2 protein protects mitochondria from being damaged and prevents apoptosis, the excess of BCL-2 in FL cells accounts for their prolonged survival [17,18]. In CLL, there are no universal mechanisms capable of justifying increased cell proliferation and/or faulty apoptosis based on defined gene lesions. For example, del13q-, the most common cytogenetic alteration in CLL [19], which causes the deletion of the genes encoding for the regulatory miR-15 and miR-16 and the consequent upregulation of *bcl-2* expression [18], is present in only approximately 50% of CLL cases and not all the cells of the leukemic clones display this alteration. Lesions which cause defective apoptosis, such as del17p-, resulting in the deletion of the *TP53* locus or inactivating *TP53* gene mutations, are detected in a relative minority of CLL cases, frequently at advanced stages and again in only a proportion of the cells of a clone [20,21]. Two other relatively common cytogenetic lesions, del11q- and trisomy 12, are observed in a minority of cases and in a fraction of the cells of single cases [22]; the same can be said of lesions such as mutations affecting the *SFB31*, *NOTCH1* or *BIRC3* genes [23,24]. Evidence has been collected supporting a germ-line predisposition for CLL, which includes the prevalence of the disease in individuals of European ancestry; the occurrence of the disease in families; and the observation that relatives of patients with CLL may develop CLL or monoclonal B cell lymphocytosis (MBL), which are considered the immediate precursors of CLL, more frequently than other individuals [25,26,27,28,29,30,31]. These observations have promoted genome-wide association studies aimed at elucidating this predisposition at the gene level. However, this investigation has led so far to the identification of single nucleotide polymorphisms (SNPs) associated with a low CLL risk, distributed over 30 loci, without identifying a single or a restricted group of lesions that could provide a unifying pathogenetic mechanism [32,33,34,35].

Although it cannot be excluded that these gene lesions may be discovered, particularly with the increasing number of whole-genome sequencing studies, we must consider the possibility that minor cytogenetic lesions, perhaps heterogeneous in nature, may predispose the cells to abnormally react to stimuli coming from the micro-environment, causing abnormal B cell proliferation and eventually leading to a full-blown leukemia. These interactions may involve cells from the micro-environment in which CLL develops and expands. In addition, signals of the BcR may have a promoting role in the expansion of leukemic cells that are deranged from the normal mechanisms of the response to antigenic stimuli. This hypothesis was initially suggested based on differences between the normal B cell repertoire and that of CLL deduced from studies of many leukemic clones, indicating the selection of certain BcR specificities, capable of favoring leukemia clonal expansion [36]. The observation that, in many CLL cases, the antigens to which the BcR has specificity could be self-antigens [37,38] or antigenic structures from pathogens [39] placed additional emphasis on this hypothesis. Finally, the fact that certain tyrosine kinase inhibitors (TKIs), capable of blocking BcR signaling, have a beneficial therapeutic effect led to further attention being focused on the pathogenetic role of BcR [40].

In this review, we analyze the characteristic features of the CLL IG gene repertoire and indicate some of the mechanisms involving the BcR of leukemic cells which can actively participate in leukemogenesis and in the maintenance and expansion of CLL clones.

## 2. The CLL Cell BcR

### 2.1. Characterizing Features of the CLL BcR Repertoire

The DNA sequences of the IGHV-IGHD-IGHJ and of IGLV-IGLJ gene rearrangements of many CLL clones have provided information on the BcR repertoire utilized by CLL cells in comparison with that of normal circulating B cells. The CLL BcR repertoire is characterized by a skewed usage of gene segments with over/under-representation of certain IGHV, IGHD and IGHJ gene segments and with frequent associations between them that are not found in normal B cells [41,42,43,44]. For example, the IGHV1-69 gene and some of its alleles are overrepresented in CLL, as are the IGHV4-34 and the IGHV3-7 genes [41,45]; The IGHV1-69 and IGHV4-34 genes are associated with autoimmunity [46,47,48]. Genes of the IGHV3 gene family (IGHV3-30, IGHV3-30.3, IGHV3-33) are under-represented [41]. Moreover, there is a greater association between the IGHV1-69 and the IGHJ6 gene segments than that observed in normal B cells (approximately 60% vs. 30%), which explains the greater HCDR3 length of many CLL cases utilizing this gene combination [45,49].

A substantial number of CLL clones share quasi-identical BcRs [50,51,52,53,54,55], which could not occur by chance given the enormous diversity potentially generated by the recombination of the IGHV and IGLV gene segments. Data accumulated over time have led to the progressive clarification of the overall features of these quasi-identical BcRs, referred to as the stereotyped BcR, and intense collaborative work conducted by different investigators has contributed to the further clarification of this issue [56]. First, several hundreds of CLL subsets utilizing the stereotyped BcR have been identified, although those most frequently encountered subsets, also called the major stereotype subsets, are limited in number. Second, there are minimum criteria that must be fulfilled to classify two or more BcR as belonging to the same stereotype (see Table 1). Third, in any given cohort, subsets represented in over 0.2% of the sequences are defined as major, whereas those represented in less than 0.2% are defined as minor [57]. More recently, these concepts have been refined, adding the notion of satellite stereotyped BcRs [56], which includes those BcRs that would be excluded from the definition of the stereotype by the stringent requirement of an identical HCDR3 length. Fourth, CLL clones from the major subsets present light chain restriction (i.e., they utilize exclusively kappa or lambda light chains) and IGLV-IGLJ rearrangements with stereotypic features such as those described for the IGHV-IGHD-IGHJ gene rearrangements [58]. These light-chain characteristics bear relevance for the classification of a given BcR within a stereotype subset. Based on their features, stereotyped BcRs could be defined as part of the public antibody repertoire, although the definition of public sequences should be kept separate from that of stereotypes since for public sequences found in experimental animal models, and in some human pathological conditions, more stringent similarity criteria are generally adopted [59,60].

With the above criteria, the European Cooperative Initiative on CLL, or ERIC, determined that patients with stereotyped subsets of any type (major and minor) peaked at 41%, and those with major stereotypes exhibited a cumulative frequency of 13.5%, in a cohort of approximately 30,000 cases [56]. Minor differences in the definition of stereotypes and the usage of different algorithms for sequence data processing in other studies may yield differences in the results obtained [61].

The discovery of the frequent usage of a stereotyped BcR by CLL cells has provided relevant information. Stereotypes, by binding to specific ligands, can facilitate the expansion of certain clones which become the target of transforming events during leukemogenesis and can promote the clonal expansion of full-blown leukemic clones. Notably, groups of CLL cases sharing the same major stereotype subsets have remarkably similar clinical courses and outcomes, a finding that reinforces the notion that BcR simulation is crucial in promoting the rate of clonal expansion [55,62,63].

### 2.2. The Issue of the Comparison between the CLL and the Normal B Cell Repertoire

Based on the data discussed above, the BcR repertoire of CLL may be different from that of normal B cells because of the presence of a stereotyped sequence and of a preferential usage of certain gene segments. Before endorsing such a statement, there are two issues to be discussed, namely, what is a BcR repertoire that can be considered normal for a CLL patient cohort, and what is the normal comparator B cell to be used for repertoire analysis?

Starting with the first point, there are differences in the BcR repertoires of U-and M-CLL, since the former group utilizes the stereotyped sequences more frequently [56] and since the most frequently utilized gene segments are in an unmutated status, for example, in the case of the VH1-69 gene [43]. Therefore, the relative proportion of U and M cases present in a cohort can substantially influence the repertoire analysis. Moreover, U-CLL cases are likely more numerous in the cohorts including cases at advanced clinical stages, given the tendency of U-CLL to progress more rapidly. For example, in the approximately 30,000 cases analyzed by ERIC [56], the proportions of M- and U-CLL were 54.1% vs. 45.9%, respectively, as expected in a cohort including cases at advanced stages. This is different from the proportions observed in cohorts at early clinical stages, where these proportions were 65–70% and 30–35%, respectively [64]. However, this consideration awaits confirmation based on cohorts that have been followed longitudinally. The selection of U-CLL cases with disease progression may find further exemplification in MBL, considered a progenitor of full-blown CLL. MBLs are represented by monoclonal accumulations of circulating B cells at levels lower than those required for CLL diagnosis (>5000 clonal B cells/microL) and are subdivided further into low-count MBL (<500 B cells/microL) and high-count or clinical MBL (>500 B cells/microL). Phenotypically, the cells from most MBLs of all types are similar to CLL cells, except for rare cases of unknown clinico-biological significance [26,27]. Progression from clinical MBL to CLL is observed in 1–5% cases/year, whereas the progression of low-count MBL is documented only in familial CLL [65]. The low proportion of progressing cases is not surprising, given the documented long persistence of MBL before transformation into full-blown CLL [66]. Clinical MBL can be grouped into U and M cases, and these share the same cytogenetic lesions, gene expression and miRNA profiles as CLL, although the proportion of U cases and of cases with unfavorable cytogenetic lesions (seen predominantly in U-CLL) is similar to that found in early CLL [67,68,69]. This is consistent with the notion of a selection of cases with unfavorable prognostic markers with disease progression. The BcR repertoire of low-count MBL is more similar to that of normal B cells than to that of CLL [70]. This feature, the paucity of U cases in this group and the rare transformation into CLL have led to the suggestion that low-count MBL cases are unrelated to CLL [3]. However, an alternative explanation could be that the rare cases resembling aggressive CLL progress more rapidly to full-blown CLL without accumulating in this group. Consequently, the overall features of low-count MBL are those of the indolent CLL with a good prognosis.

In conclusion, CLL repertoire analyses must take into consideration the case-mix of the cohort studied.

The issue of the normal B cell comparator for repertoire studies is relevant to assessing the cells and the mechanisms of origin of CLL. The observation of a skewed repertoire and of a stereotyped BcR may suggest that in CLL there is a selection of certain BcR specificities. Alternatively, CLL cells could originate from a B cell subset with a BcR repertoire similar to that of CLL, raising the question of which B cell subset represents the correct comparator for repertoire analyses. This issue has been amenable to investigation only recently, owing to the availability of NGS databases. B cells from the spleen and peripheral blood were fractionated according to a phenotype (assessed with mAbs) which reflects their anatomical location (e.g., follicular mantle, germinal center, marginal zone B cells, etc.) or to particular cell features (e.g., CD27^+^ memory, switch-memory, sIgM^+^ only, sIgM^high^, sIgD^low^ B cells, etc.). All these subsets were analyzed for the presence and distribution of CLL-like stereotyped IG sequences (CLS-IG). Since CLS-IG are tumor-related sequences, the CLS-IG frequency and distribution would have given a measure of the correlations between the CLL repertoire and that of the normal B cell subsets [71,72]. CLS-IGs were observed in virtually all the cellular subsets analyzed and were more numerous in those with more U than M sequences because of the higher representation of stereotypes in U sequences. Nevertheless, the CLS-IG distribution was very different from that of CLL, since certain stereotypes that were present in large amounts in CLL were not so numerous in the different B cell subset, and vice-versa. In addition, the strong restriction observed in CLL stereotypes for kappa or lambda light-chain utilization was not observed in the CLS-IG from any of the normal B cell subsets analyzed. An additional preliminary finding was that the skewed repertoire usage typical of CLL could not be identified for any of the B cell subset populations (Bagnara et al., in preparation). Therefore, it appears that normal B cell subsets sharing repertoire characteristics with CLL do not exist, or, if they exist, are confined to certain peripheral lymphoid tissues, such as the gut or lung, which were not analyzed. However, the rare extra-nodal localization of CLL makes this possibility rather unlikely.

A special problem is represented by the repertoire of normal B cells expressing surface CD5. In the past, these were considered as the cells of origin of CLL, based mainly on phenotype considerations [73,74] and on observations in mice, in which they represent a homogenous cell subset [75]. In mice, CD5^+^ B cells are numerous in the peritoneal cavity (albeit with differences in the various strains of mice); express a particular BcR repertoire; produce polyreactive antibodies (i.e., antibodies reacting with several antigens including self-antigens) mainly of the IgM isotype, as many CLL clones do; and give origin to lymphoproliferative disorders in certain experimental models (see below). Because of the features of CD5^+^ B cells, murine B cells are subdivided into B1 B cells (i.e., the CD5^+^ B cell population), which are capable of T-cell-independent responses and of self-replenishing, and B2 B cells, which are capable of T-cell-dependent responses, of antibody affinity maturation and of isotype switching for the production of IgG and IgA antibodies.

A convincing demonstration that CD5^+^ B cells represent a homogeneous subpopulation in humans is lacking and most evidence supports the contrary. This issue has been already dealt with in a preceding review and here we summarize some relevant notions [76]. Circulating human CD5^+^ cells are heterogeneous; most of them are naïve B cells, but there is also a minority of IgM^−^ and isotype-switched memory B cells. In the spleen and tonsils, CD5^+^ B cells are composed primarily of naïve B cells with the phenotype of follicular mantle cells but contain B cells of different anatomical origin such as marginal-zone B cells. Moreover, unpublished observations from our laboratory confirm the heterogeneity of CD5^+^ B cells from the peritoneal cavity. A unifying feature of all CD5^+^ cells is that they appear to be activated cells as demonstrated by the expression of additional activation markers, a concept which is in line with the notion that the activation of cells from different subsets leads to CD5 expression. Although initial studies with Sanger sequencing methodologies disclosed repertoire analogies of circulating CD5^+^ B cells with CLL [77], this was not confirmed by NGS investigations, particularly when the CLS-IG distribution was analyzed [72]. This distribution was found to be similar to that of the other B cell subpopulations, except for a marginally higher proportion of CLS-IG, which was correlated with the higher representation of U sequences.

Taken together, the above observations suggest that the process of leukemogenesis is characterized by selection in the repertoire of CLL clones.

## 3. Mechanisms of CLL Cell Stimulation via BcR

Lymphoproliferative disorders of mature T and B cells are generally believed to originate from cells that have completed their maturation process. This is unlike what has been observed for most cancers, in which the transforming events occur primarily in the stem cells, which, because of these events, become cancer stem cells, capable of proliferating and of differentiating in part into mature neoplastic cells, with low proliferative capacities [78]. This genesis is possible since mature T and B cells can be recruited into the cell cycle via external signals and propagate the transforming events to their progeny, as stem cells of other tissues do. The stimulation of mature T and B cells by antigens may represent one of the signals which facilitates the process of malignant transformation. Gastric lymphomas are accompanied by *Helicobacter pylori* (HP) infection, which promotes the formation of lymphoid tissue in the gastric mucosa and the proliferation of B cells that already carry transforming mutations or which cause the accumulation of transforming events while proliferating [79]. Antibiotic treatment results in a regression of the lymphoid gastric lesions, concomitant with the eradication of the HP infection in the early disease stages, whereas it is ineffective or partially effective at later stages, indicating the role of additional transforming events in disease progression [80]. In lymphomas arising in patients with hepatitis C virus (HCV) infection, the BcR of the malignant clone frequently has specificity for HCV epitopes and treatment of the viral infection can result in lymphoma regression [81,82]. In CLL, the analysis of the repertoire indicates the involvement of BcR in promoting clonal expansion. Here, we analyze the mechanisms involved in this process.

### 3.1. BcR Signaling in CLL

Two types of signals can be delivered by BcRs, i.e., tonic and active signals, according to the current terminology [83], with different functions relevant to leukemogenesis.

#### 3.1.1. Tonic Signals

These signals allow the survival of mature B cells in vivo and an active engagement of the BcR with a specific antigen epitope is not required. This conclusion stems from the observation on the conditioned inactivation of surface IgM or of CD79a, the BcR accessory chain, which is part of a disulfide-linked heterodimer (CD79a/CD79b) subunit capable of mediating BcR membrane expression, signal transduction and receptor internalization [84,85]. In mice, this inactivation causes a progressive and prolonged reduction in the total number of circulating B cells over a period of weeks. The mechanisms of the delivery of tonic signals are not fully understood, although it has been proposed that BcR is expressed at the cell surface in the form of stable auto-inhibited oligomers which re-organize into clusters in the presence of the appropriate BcR ligand, which are capable of driving cell activation [86,87]. Stable oligomers can still deliver tonic signals by activating the PI3K pathway which is one of the pathways that is dependent on the engagement of BcR (see Figure 1). The activation of this pathway may imply Syk activation or may be operated directly via CD79b activation [88]. The requirement of tonic signals explains why the cells from most lymphoproliferative disorders of mature B cells express fully assembled surface IgM molecules, the Ig isotype most suitable to deliver tonic signals [83]. The majority of CLL clones express surface IgM, irrespective of whether they belong to the U- or M-CLL group. This finding is somewhat unexpected since M-CLL cells are likely to have passed through germinal centers where somatic IGHV hypermutation (SHM) and isotype switching take place, although it may relate to the capacity of surface IgM to deliver more efficient tonic signals than IgG or IgA [83]. Although intra-clonal isotype switching from IgM to IgG or IgA is observed in many CLL cases, it is confined to a minor sub-clonal component, with a few exceptions, as it may not offer a special survival advantage to the leukemic cells [89,90]. Certain CLL cases express stereotypes invariably connected with an IgG isotype, such as subset #4 or #8 stereotypes, suggesting that tonic signals are substituted for by other more advantageous signals in these clones [56]. However, unlike normal resting B lymphocytes, CLL cells frequently have some IgM clusters on their surface, suggesting that tonic signals are delivered together with active signals in many circumstances, as outlined below [91].

#### 3.1.2. Active Signals

These have the function of promoting cell proliferation and clonal expansion and include classic antigenic stimulation (extrinsic BcR engagement) (Figure 1) and BcR self-recognition (intrinsic BcR engagement) (Figure 2). These signals can be further subdivided as follows.

(i)*Stimulation by self-antigens.* The presence of frequent auto-immune manifestations, such as auto-immune hemolytic anemia or thrombocytopenia, suggested a connection between CLL and auto-immunity since the early studies [92]. This notion was substantiated more recently by the observation that a considerable number of CLL clones expressed a BcR characterized by poly-reactivity, a definition indicating that each monoclonal antibody could react with low affinity with a variety of different (auto)antigens, including platelets, aggregated IgG, nuclear antigens, double-strand (ds) and single-strand (ss) DNA, insulin, etc. Antibodies with these features are found among the “natural antibodies”, a family of antibodies mostly of the IgM isotype, of all mammalian species, representing one of the first lines of defense against assaulting pathogens [37,38]. These concepts were further refined by showing that U-CLL clones produce polyreactive IG very frequently, whereas this occurs rarely for M-CLL clones [93]. However, autoantibodies from patients with auto-immune manifestations, observed in both U- and M-CLL patients, are not produced by leukemic cells, since they are of the IgG isotype, they utilize both K and lambda light chain types and they are polyclonal [94]. Therefore, auto-immunity is not directly caused by the leukemic clone, although models in which leukemia and autoimmunity are part of the same pathogenetic process can be proposed, as we shall discuss later. Rare patients with cold agglutinin disease or cryoglobulinemia and CLL represent a notable exception. In these conditions, leukemic cells produce monoclonal low-affinity auto-antibodies to red cells or to the Fc portion of IgG [48,94,95], a feature consistent with the limited capacity of CLL cells to mature into plasma-cell-secreting antibodies [96]. Additional information came from the observation that poly-reactive antibodies often recognize antigens at the surface of apoptotic cells. Although normally located intracellularly, certain self-antigens can be expressed at the cell surface, when apoptosis is activated, and can be modified by metabolic processes, such as oxidation associated with apoptosis [97,98,99]. One physiologic function of poly-reactive antibodies is the clearance of apoptotic cells [100]. The expression of a BcR with polyreactive features, capable of recognizing apoptosis-related antigens, may become instrumental in promoting CLL clonal expansion, particularly in the presence of abundant CLL cell apoptosis. A substantial proportion of IG molecules cloned from or secreted by CLL cells react with intracellular proteins such as vimentin, tubulin and filamin B exposed at the cell surface following the induction of apoptosis [99]. Additional proteins to which these IGs have reactivity are those with which the sera of systemic lupus erythematosus (SLE) patients have reactivity, including Sm, snRNPA, Ku and other molecules which are also recognized both in the native or oxidized form by IG from the sera of SLE patients [97]. IG molecules with these reactivities are produced predominantly by U-CLL cases and many have stereotypic features, particularly those encoded for by the VH1-69 genes [97,98,99]. However, the absence of IGHV mutations and/or the utilization of a given stereotype do not classify these BcRs as specific for apoptosis-related antigens, since several of them fail to show any reactivity and different reactivities have been detected for the different BcR or BcR families investigated.(ii)*Stimulation by microbial antigens.* In the absence of a clinically evident infection, such as the HP infection in gastric lymphoma, it is difficult to determine whether the BcRs of CLL cells may have specificity for antigens of a given pathogen. Nevertheless, researchers were able to trace a BcR with specificity for certain microorganisms in CLL. Hoogeboon and colleagues [39] analyzed 82 CLL patients, whose cells expressed an IGHV3-7-encoded BcR. The choice of this cohort was suggested by the reported over-representation of this gene in CLL and by the observation of a frequent SHM in these CLL clones, indicating antigenic stimulation and passage through germinal centers. A further selection within the cohort led to the choice of four patients in whom the BcR was characterized by a very short HCDR3 sequence of 5–6 amino acids (aa) instead of the canonical 15 aa. These BcRs were characterized by the utilization of nearly identical IGKV2-24-encoded Ig light chains and for sharing a glutamic acid at position 106 of HCDR3, which was not detected in any of the other sequences utilizing the IGHV3-7 gene. Cloning of genes and the expression of fully assembled IgM molecules led to the observation that the antibody bound 4/33 commensal yeast species and presented a specific binding to the β-(1,6)-glucan of the yeast. The substitution of the glutamic acid at position 106 of the HCDR3 via site-directed mutagenesis caused inhibition of the high-affinity binding to glucan, as did the substitution of certain aa of the short HCDR3. Finally, in vitro exposure of CLL cells with this BcR to the β-(1,6)-glucan resulted in specific cell proliferation. Together, these observations suggest a process of antigen selection, which may be more frequent than possibly thought, particularly if the stimulating antigens are carried by commensal rather than pathogenic microorganisms. Autoreactivity, described in the preceding section, and reactivity with microbial antigens may represent two aspects of the same phenomenon. Antibodies reacting with molecules exposed on the surface of apoptotic cells and/or with molecules oxidized following apoptosis show cross-reactivity for several microbial components. Therefore, this cross-reaction, which can be demonstrated in vitro, may also operate in vivo in CLL patients and certain CLL BcRs can concomitantly recognize antigens of micro-organisms and self-antigens [99].(iii)*Autonomous signaling.* This definition relates to the capacity of CLL cell BcRs to autonomously deliver activation signals to leukemic cells, as discovered in a very elegant system in vitro. Using retroviral gene transfer, BcRs from CLL clones were expressed in murine cells lacking endogenous BcR components and were thus unable to be signaled via their own BcR [101]. Positive signaling, revealed by Ca^++^ mobilization, was consistently noted following cell transfection with the BcRs from 17/17 CLL clones and not with the BcRs from other lymphoproliferative disorders including mantle cell, marginal zone and follicular lymphoma and myeloma (15/15 cases, collectively). This phenomenon was observed with the BcRs from U- and M-CLL cases and from cases expressing BcRs with/without poly-specific reactivity. Autonomous signaling also was observed with the BcRs from leukemic cell clones from mice that were transgenic for the *T cell Leukemia 1* or *tcl1* gene. These mice, obtained following the observation that U-CLL clones showed high TCL1 protein levels, expressed a *tcl1* transgene under the control of the IGHV promotor and the E*μ*-enhancer [102]. They developed a lymphoproliferative disorder characterized by the expression of CD5 and the utilization of unmutated IGHV genes, thus resembling human U-CLL [103]. Autonomous signaling was not observed when transfecting the BcRs from murine non-leukemic clones specific for several antigens, indicating that the phenomenon occurs preferentially in malignant cells [101]. Autonomous signaling can be reminiscent of that observed in other cancers, such as breast, lung or gastrointestinal tract cancers, in which activating mutations of receptors for growth factors enable the receptors to deliver signals similar to those of normal receptors following interaction with specific ligands [104,105]. Because of mutations, the receptor acquires the property of self-aggregation or dimerization or undergoes conformational changes, which can activate downstream signals (see Figure 2). However, there are no activating mutations in the BcR transducing the downstream signaling, and the system is more complex in comparison to these models. Ca^++^ mobilization is observed when the transfected BcR displays a HCDR3 motif which enables binding to a conserved epitope in the framework region 2 (FR2) of the IGHV domain of the same BcR molecule [101]. An alternative epitope for BcR autologous binding is found in FR3 [106]. Although these mechanisms have the same effect as the activating mutations of growth factor receptors present in other cancers, the underlying molecular mechanisms are different, as the recognition of a specific epitope of the “leukemic” BcR by the antigen-combining site of the same BcR expressed by the leukemic clone is required for receptor activation. In other words, autonomous signaling is intra-clonal and implies a specific interaction between BcR molecules on the surface of the same cell (or the BcRs of different cells from the same clone), given that the BcR recognizes a self-epitope (see Figure 2) [101]. A good example of the relevance of autonomous signaling is provided by a subgroup of CLL of the subset #2 stereotypes. These cases carry a single-point mutation, termed R110, at the junction between the variable and the constant region of the light chain. Although subset #2 stereotype is itself a poor prognosis indicator, R110^+^ cases display an even worse outcome. Light-chain mutation has been found to facilitate homotypic BcR-BcR interactions which result in a more robust activation of autonomous signaling [107].

Not all the types of BcR engagement occur concomitantly or operate in all CLL clones. Some of the signals depend on the BcR type. Poly-reactivity is frequent in U- and rare in M-CLL clones; hence, the BcR engagement is different in the two groups, explaining in part their differing clinical behavior. Some of the BcR engagements operate constantly during the disease, as is the case for self-antigens delivering signals to a polyreactive BcR or for autonomous signaling, whereas others are intermittent, as is the case for antigens of pathogens, which can deliver stimulatory signals during infections, although cross-reacting epitopes of apoptosis-related antigens can continue to stimulate the leukemic clone in the absence of pathogens. The presence and quantity of a given antigen may represent a limiting factor in BcR engagement, as is perhaps the case for CLL patients whose cells express a BcR with rheumatoid factor-like specificity, since aggregated IgG, in the form of immune complexes, may not be constantly available for stimulation [48]. A successful BcR engagement may require support that only particular anatomical sites can offer. The presentation of self-antigens to a polyreactive BcR may be facilitated by the cellular/molecular structure of the proliferating centers of lymphoid tissues and prevented in circulating CLL cells [108]. Autonomous signaling may not have these topographical limitations, although the differing availability of accessory cells in the various sites may impose different constraints in the outcome.

Some signals may be operative only in certain periods of the history of a leukemic clone. The BcR of M-CLL can be made to revert to the U status using site-directed mutagenesis and can regain poly-reactivity, suggesting that M clones are derived from a U progenitor B cell [93]. Although not universally accepted, as we shall discuss later, this hypothesis opens the possibility that the change from U to M features during leukemogenesis also requires a change in the type of BcR stimulation, possibly with a switch from a poly-reactivity-type to an autonomous-signaling-type BcR engagement. The transforming CLL clones could adapt the type of BcR engagement appropriate to their needs during leukemogenesis. For example, leukemogenesis could be promoted initially by cell proliferation induced by a pathogen that also induces somatic hyper-mutation (SHM). When infection ends, SHM could be selected based on autonomous signaling rather than the affinity maturation requirements. In M-CLL, the quality/quantity of IGHV mutations does not affect the clinical course, possibly suggesting that SHM is driven more by autonomous signaling than by affinity maturation needs [109,110]. The differences in the strength of autonomous signaling reported in clones expressing different stereotyped BcRs may reflect a similar selection process [111].

The stimuli illustrated above provide a sufficient array of mechanisms whereby the BcR can keep CLL cells in an activated status in vivo. This condition is documented by the expression of activation markers detected at different extents depending upon the subgroup of origin (e.g., U- or M-CLL) and by the presence in unmanipulated CLL cells of a number of phosphorylated molecules belonging to the BcR-dependent signal transduction pathway [112,113].

A final note of caution concerns the fact that most BcR engagements proposed for the survival/expansion of CLL cells have been deduced from in vitro observations and the results may not reflect the real situation in vivo, although some of the mechanisms have found a further demonstration in vivo in murine models of lymphoproliferative diseases.

### 3.2. Outcome of BcR Engagement

The outcome of BcR engagement is heterogeneous, ranging from activation to anergy, and from proliferation to apoptosis, and can be modulated by the microenvironment. Several aspects of these regulatory mechanisms are analyzed below.

#### 3.2.1. Identification of the Proliferating Cell Fraction of the CLL Clone

Since the early studies on CLL cell surface markers, it was clear that CLL clones with high numbers of CD38^+^ cells responded to surface BcR stimulation, whereas those with low CD38^+^ cell numbers could be classified as low responders or anergic [114,115]. The same was observed with CLL clones classified as ZAP-70^high^ and ZAP-70^low^ or U and M, respectively [116,117]. BcR cross-linking was achieved by exposing the cells to a divalent anti-IgM antibody and cell stimulation was measured via Ca^++^ mobilization and/or protein tyrosine phosphorylation. These data could suggest that some CLL clones only responded to stimulation via BcR, whereas the expansion of the others followed different pathways. Intra-clonal fractionation of cells according to CD38 expression indicated a different interpretation, since the CD38^+^ cells only proved capable of a response via BcR stimulation, whereas the CD38^−^ cell fractions were invariably anergic and this was observed irrespectively of whether the two fractions were obtained from clones classified as CD38^high^ or CD38^low^ [118]. In line with this result was the observation that CD38^+^ cells were comprised of more Ki67- and ZAP-70-expressing cells and had higher telomerase activity than CD38^−^ cells from the same clone, suggesting that CD38^+^ cells were those with the highest proliferating capacity [119]. This finding was confirmed by observations with deuterated water labeling in vivo, demonstrating that the circulating CD38^+^ cells were enriched for recently divided cells [11,120].

Tests with deuterium labeling in vivo and gene expression profile (GEP) methodologies on lymphocytes from different sites showed that the proliferation of CLL cells takes place predominantly in the peripheral lymphoid tissues [121,122]. Moreover, immunohistochemical observations indicated that the proliferating centers in peripheral lymphoid tissues were the sites of the highest leukemic cell proliferation, given the enrichment in the expression of Ki67^+^ leukemic cells co-expressing activation markers [108]. Here, the CLL cells have the opportunity of being in close contact with accessory cells such as macrophages, nurse-like cells (NLC) and T cells and with the cytokines that they release [123,124,125,126]. Thus, the combination of effective BcR engagement, through one or more of the mechanisms described above, facilitated by the environment of proliferating centers, together with the assistance of a variety of accessory cells, appears to promote the process of clonal expansion. Leukemic cells in lymphoid tissues are characterized by high CXCR4 expression, which facilitates their homing by interacting with its ligand, CXCL12, on NLC [124,126]. In addition, chemokines such as CCL3 and CCL4 released by the leukemic cells exert a chemo-attractant function for lymphocytes and monocytes with specific receptors (CCR1 and CCR5) [125]. The activation of CLL cells in proliferating centers can result in the lowering of the CXCR4 levels, which facilitates the cells’ exit from proliferating centers into the circulation, where they continue to express activation markers at an extent proportional to the activation in the proliferating centers [127]. CD5, an activation marker capable of regulating BcR activity, is also upregulated [128]. However, the diminished chances of stimulation in the circulation contribute to a decrease in the activation of leukemic cells and to lower levels of CD5, while increasing CXCR4 levels. This makes the cells ready to re-enter the peripheral lymphoid organs and to initiate a new cycle of proliferation [129]. The levels of CD5 and CXCR4 expression have been used to define three groups of CLL cells: the CD5^high^ and CXCR4^low^ cells, which are recently divided cells that express high levels of activation markers including CD38; the CD5^low^ and CXCR4^high^ cells, resting CLL cells which are prone to re-enter lymphoid tissues; and the CD5^intermediate^ CXCR4^intermediate^ cells, which are in between the two other categories and are likely cells at various activation states and/or on their way to becoming quiescent [130,131]. These distinct groups can be identified in both M- and U-CLL clones, confirming that both subsets have proliferating compartments, as anticipated by the initial fractionation studies conducted using surface CD38. Observations on circulating CLL cells, fractionated according to the density of sIgM, provided another means of cell subdivision. Four subgroups were identified and named SG1–4, with increasing sIgM density from SG1 to SG4. Cells from SG1 had the highest levels of activation markers (CD25 and CD38) and the lowest capacity to be stimulated by sIgM cross-linking, determined based on protein tyrosine phosphorylation [132]. The other SGs progressively lost their activation status and regained the capacity of expressing sIgM, together with a more efficient response to sIgM cross-linking. The cells from SG4 also had the highest CXCR4 levels and were classified as “dangerous cells” based on their presumed capacity to re-enter lymphoid tissues and to re-initiate a proliferative cycle, feeding the proliferating compartment of CLL cells. Again, these distinct cell fractions were detected in both U- and M-CLL clones, albeit in different proportions. The two models present differences (not unexpected in view of the complexity of the fractionation procedures) related to the definition of the cell subsets that were responsive to BcR engagement. Nevertheless, these studies have the merit of disclosing heterogeneity within CLL clones, with cells that represent the proliferating compartment and with others that form a sort of reservoir of “stem cells” capable of re-entering the cycle to make up for the continuous cell loss. Figure 3 presents a hypothetical model of CLL cell turn-over based on the data described in this section.

#### 3.2.2. Choice between Proliferation and Apoptosis following BcR Stimulation

The stimulation of unmanipulated cells from U-CLL cases with divalent anti IgM antibodies results in survival/proliferation or apoptosis in vitro, depending on the stimulus and the presence/absence of accessory cells and/or cytokines [114,116,117,133,134]. Exposure to anti-IgM-coated beads facilitates proliferation, whereas soluble antibodies favor apoptosis. In our own experience, testing highly purified CLL cells favors apoptosis, whereas the presence of minor contaminants of T cells and/or monocytes favors proliferation, indicating a pro-survival role for these accessory cells, a finding in line with the observation that purified CLL cells, particularly from U-cases, do not survive in culture for long, unless accessory cells, such as NLCs, stromal cells, fibroblasts, activated T cells and macrophages, are added [135,136,137,138,139]. Collectively, CLL cells appear to be in a precarious balance between survival and death in vitro and this condition is exacerbated by BcR stimulation alone. This situation is likely to occur in vivo as well, given the abundant apoptosis observed in patients’ circulation. How the balance between survival and apoptosis is regulated is not entirely clear. *Bcl-2,* which is not mutated or translocated in CLL cells [24], is generally upregulated. This finding is in part related to del13q-, which characterizes many leukemic clones and causes the deletion of the locus encoding miR-15 and miR-16, with the function of negatively regulating *bcl-2* expression, and in part due to the activation status of CLL cells, leading to the upregulation of anti-apoptotic mechanisms [140,141,142,143]. However, many pro-apoptotic molecules, particularly those of the BH3 family alone, are also upregulated, counterbalancing the anti-apoptotic effect of the molecules of the BCL-2 family [144]. This hypothesis also is supported by the therapeutic effects observed in CLL patients with BH3 mimics, which, by inhibiting the anti-apoptotic effects of the *bcl-2* family molecules, swing the balance toward the induction of apoptosis [145]. *c-myc,* which is not mutated or translocated in CLL [24,146,147], may also contribute to the apoptosis of CLL cells since it is frequently upregulated in CLL cells, possibly because of stimulation via BcR and/or other accessory cells in vivo [148]. In addition, *c-myc* has been found to be upregulated in different murine models of CLL [149,150,151]. It has been known for a long time that *c-myc* upregulation renders cells prone to apoptosis unless anti-apoptotic mechanisms are activated [152]. For example, lymphoblastoid cell lines transfected with an upregulated *c-myc* undergo apoptosis unless signals from CD40/CD40L interactions at the cell surface upregulate the cells’ anti-apoptotic mechanisms [153]. Likewise, a similar, *c-myc*-controlled imbalance is created in the centro-blasts in the germinal centers (GC), where *c-myc* upregulation following antigen stimulation renders the cell prone to apoptosis. Apoptosis is prevented by interactions with T cells facilitated by antigens, favoring high-affinity maturation [154,155,156]. Therefore, *c-myc* upregulation in CLL cells may favor apoptosis but may be counterbalanced by microenvironmental signals. Finally, CLL cells have higher-than-normal levels of radical oxygen species (ROS), possibly related to BcR engagement and/or other stimuli. These ROS act as secondary messengers, driving the leukemic cells towards proliferation or apoptosis depending upon the balance between the ROS concentration and the cell’s ability to prevent ROS damage [157,158,159].

The above mechanisms investigated in vitro reveal that the outcome of cell stimulation via BcR is largely dependent on the additional help that leukemic cells receive from accessory cells and cytokines and reinforce the view that CLL clonal expansion depends on close stromal interactions. BcR engagement may possibly serve to focus this help on CLL cells.

#### 3.2.3. Tolerance of CLL Cells

A considerable proportion of CLL clones show a low response to sIgM cross-linking in vitro, measured based on increased protein tyrosine phosphorylation compared to the background level, or by determining Ca^++^ mobilization, or the biological effect of stimulation. The cells with these features, defined as anergic cells, represent most circulating cells from the CD38^−^, ZAP-70^−^ and M IGHV CLL clones. For the sake of clarity, we speak of anergic cells within clones, rather than anergic clones, since apparently anergic clones also comprise a minority of CLL cells that are potentially capable of proliferating following BcR engagement, as already discussed. Anergic cells exhibit low sIgM, almost normal sIgD, elevated levels of spontaneously phosphorylated Syk and Lyn in addition to ERK1/2 and MEK1/2, and increased NF-AT phosphorylation in the absence of AKT activation [160,161,162]. Excessive Lyn phosphorylation causes the phosphorylation of the immunoreceptor tyrosine inhibitory motif of surface CD5 with subsequent docking and activation of SHP-1, with an inhibitory effect on apoptosis and on any further cell stimulation [163]. Inhibition of the phosphorylation of the molecules described above results in a restoration of BcR responsiveness, generally followed by apoptosis [160,161,162,163]. The features of the tolerant CLL are reminiscent of the B cells of mice that were transgenic for a given BcR and engineered to produce the antigen to which the BcR had specificity [164,165]. These B cells failed to respond to the “transgenic” antigen when challenged with it in vitro. The same mechanism is likely to operate in regard to tolerance to self-antigens in humans. IgM^−/^IgD^+^-tolerant cells that are potentially capable of producing self-reactive antibodies have been described in normal human circulating B cells [166]. It is not clear why tolerance is present in part of a CLL clone, although it could be ascribed to the continuous stimulation of the cells via their BcR in vivo, perhaps in the absence of the appropriate accessory cells. This tolerance can be overcome in vitro, for example, via stimulation with CpG (which engages TLR-9) or by exposing the leukemic cells to CD40L-bearing cells or via cross-ligation of surface IgD [57,161,165]. The breaking of tolerance is documented by the new capacity of the cells to respond to stimuli and subsequently proliferate. Although comparisons between in vitro and in vivo testing are always difficult, the data suggest that the tolerance of leukemic cells is more transient than that observed in normal human B cells or in the B cells from experimental animal models to self-antigens. It is possible that smaller groups of anergic cells also exist in the circulation in U-CLL cases, which may be masked by the overwhelming presence of cells responding to BcR stimulation in vitro. If this is the case, the induction of transient tolerance may operate in both U- and M-CLL cases, albeit in different proportions, and tolerant CLL B cells may constitute a reservoir of cells that are inducible to cycling when required for clonal expansion.

#### 3.2.4. The Function of BcR Other Than IgM

The large majority of CLL clones express both sIgM and sIgD, which share the same L chain type and the same antibody combining site. The two isotypes are assumed to have overlapping functions, although sIgD, because of its more extended hinge region (i.e., the amino-acid sequence located between the antigen binding, or Fab portion, and the tail, or Fc portion, of the molecule), is believed to have a superior flexibility and capacity for binding certain antigens with multiple epitopes compared to the more rigid sIgM, which binds antigens with fewer repeated epitopes [167,168]. These differences present a wide range of antigen binding options for B cells, offering superior defense from pathogens.

The sIgM and sIgD density is lower in CLL than in normal B cells, a phenomenon related to in vivo endocytosis caused by chronic BcR stimulation, suggesting the participation of both isotypes to cell stimulation, although with noticeable differences. For example, tests with deuterated water in vivo demonstrated that the CLL cell birth rate (BR) correlates with the sIgM and not the sIgD density of circulating CLL cells [10,169]. This is in line with the observation that patients with the highest sIgM levels in their circulating cells have a more aggressive disease, a correlation not found for sIgD [10,132]. In normal B cells, sIgM is organized in more numerous high-density regions than sIgD [170]. Furthermore, sIgM and sIgD have different topographic relationships with membrane CD19 and CD20 co-receptor molecules during the activation of normal B cells [171]. If demonstrated for CLL cells as well, this would further stress the complementary role of the two isotypes [169,172,173]. Finally, while sIgM is efficient in eliciting autonomous signaling, sIgD is not, as shown by in vitro tests and observations in *tcl1* transgenic mice that were unable to express sIgM [169]. This possibly might occur because the recognition of internal BcR epitopes by the BcR takes place only when the epitopes of interest are on IgM and not on IgD. Alternatively, the flexibility of IgD may not allow the transduction of an effective signal when the BcR is engaged by itself [168,169,173].

The above data could indicate a predominant role of sIgM in facilitating clonal expansion, with a note of caution since most findings relate to circulating CLL cells and the correlations observed are not of univocal interpretation. Nevertheless, it has been possible to observe stimulation of CLL cells via sIgD cross-linking in vitro, although the downstream signals are more transient and less effective than those of sIgM [172,174]. These signals may be able to rescue CLL cells from anergy [161] and can drive a fraction of CLL cells into plasma cell differentiation, rather than proliferation [174], thus explaining the presence of monoclonal IgM-secreting cells in many CLL clones [96,175]. The secreted monoclonal protein may interfere with autonomous CLL signaling by preventing the interaction between the BcR and the corresponding sIgM target. An alternative and not mutually exclusive explanation may be that the secreted IgM forms aggregates with the help of the BcR itself and engages the Fc*μ* receptor of the CLL cells, providing additional regulatory signals [176,177]. However, the molecular mechanisms underlying this regulation must be clarified.

The stimulatory function of sIgG or sIgA, expressed by a minority of CLL cases, does not seem to differ from that of sIgM. The reason for which sIgM expression may be predominant in CLL and other lymphoproliferative disorders has already been discussed above, although it remains to be clarified why clones expressing sIgG or sIgA have a selective advantage in certain circumstances. Clones expressing certain stereotypes may provide some clues. For example, subset #4 and subset #8 stereotypes are typically associated with an IgG isotype [56]. A selection process favoring the intra-clonal emergence of sIgG-bearing cells with these stereotypes over those with sIgM molecules and the same stereotype has been reported [178]. A combination of a given stereotype with a particular isotype would be advantageous for stimulation involving certain antigens or for autonomous signaling (as apparently occurs for subset #4 stereotypes) [111], although more observations are needed.

### 3.3. Inhibition of BcR-Dependent Tyrosine Kinases (TK)

The evidence that stimulation via BcR is one of the driving forces for CLL cell expansion promoted the screening of a variety of TKIs capable of blocking BcR-dependent signaling, with potential therapeutic value. The analysis of this research effort goes beyond the scope of this article, and we refer to the excellent ad hoc reviews presented in [2,40,179,180]. However, it is perhaps useful to add a few notes on the results obtained with widely used TKIs, such as ibrutinib and idelalisib, which covalently target the Bruton TK or the PI3K-delta isoform, respectively. These TKs are involved in the early phases of BcR signaling (see Figure 1) and participate in the signaling of other receptors essential for cell trafficking and homing. As is apparent from Figure 1, several TKs are involved in BcR signaling, some of which are located in different arms of the pathway, as is the case for Bruton TK and PI3K, raising the possibility that, in the future, different TKs can be simultaneously targeted with different specific drugs, although this strategy will have to be confronted with the issue of the amplification of adverse events [181].

Treatment with TKIs is followed by lymphoid tumor shrinking and progressive disease remission. This effect is more rapid in U-CLL cases, although clinical remission is also observed in most M-CLL cases [182,183,184,185], which is consistent with the higher hierarchical role of the BcR in promoting U-CLL cell expansion, but which also confirms a role for BcR in M-CLL. TKIs induce peripheral lymphocytosis at the start of therapy because of the redistribution of leukemic cells from lymphoid compartments to the circulation, due to many effects, including the inhibition of BcR signaling induced by exogenous or self-antigens, of autonomous signaling and of signaling from other surface structures, such as CXCR4, that contribute to the retaining of CLL cells in tissues. Notably, lymphocytosis observed following TKI administration is established more rapidly in U-CLL than in M-CLL, although it persists for a longer time in M-CLL [183]. This is consistent with the more effective role of BcR in U-CLL expansion but is also possibly related to a recently described effect of myristoylated alanine-rich C-kinase substrate or MARCKS, which is more expressed in M-CLL cells than in U-CLL cells [186]. Since the role of MARCKS is that of partially inhibiting BcR signaling by reducing BcR clustering, this observation contributes to explaining why M-CLL, receiving less effective stimulation via BcR, may be less susceptible to TKI.

When moving to the circulation under the effect of TKIs, the leukemic cells meet with conditions which are not optimal for a survival/proliferation and progressively die. Circulating CLL cells from patients treated with ibrutinib present the upregulation of surface IgM (compared to pre-treatment levels), which is not followed by an equal up-regulation of surface IgD levels [186,187,188]. Disengagement of the surface BcR caused by treatment is likely to reinstate the pre-stimulation conditions of CLL cells and is particularly evident at the level of surface IgM, the most frequently engaged isotype [187]. CD20, another molecule participating in BcR stimulation, is downmodulated following treatment with TKIs, which also induce a downregulation of CXCR4, that is responsible for leukemic/stromal cell interactions in tissues. Downregulation of CXCR4 facilitate the cells’ exit from the circulation and affects the production of CXCL12 by stromal cells. Reduced signaling by CXCL12 is responsible for the lowering of CD20 [189].

The effectiveness of TKIs is reinforced by the observation of disease re-expansion in patients who have developed BTK or PLCγ2 gene mutations and acquired TKI resistance. The former mutations prevent the binding of TKIs, such as ibrutinib, to the Bruton TK, whereas the latter confer independence from Bruton TK to the BcR’s signaling [183,190,191].

## 4. Stimulation of CLL Cells via Surface Structures Different from the BcR

The stimulation of CLL cells via the BcR can be modulated by additional signals that are capable of determining the outcome; for example, sIgM cross-linking is often followed by cell apoptosis or anergy in vitro, although the addition of IL4, IL2, IL10 or IL15 may induce some cell proliferation and prolong cell survival [114,192,193]. A similar modulating effect can be exerted by the presence of accessory cells including T cells, macrophages or NLC and/or by agents such as microbial DNA, viral RNA or unmethylated CpG-motif-containing oligodeoxynucleotide (CpG-OND), which are capable of binding structurally conserved molecules, as part of native immunity, such as the Toll-like receptors (TLRs) [194,195,196]. Finally, factors such as B-cell-activating factor (BAFF), proliferation-inducing ligand (APRIL) and stroma-cell-derived factor-1 𝛼 (SDF-1 𝛼) may contribute to the survival and proliferation of leukemic cells [197]. Notably, some of the pathways involved in these modulating signals (e.g., CD40/CD40L interactions, TLR stimulation and cytokine signaling) overlap with BcR signaling in certain intracellular key check-points, such as Bruton TK, although their cross regulation is not yet fully understood [198,199,200,201]. It is possible that following interaction with accessory cells, especially those expressing CD40L, leukemic cells can proliferate in the absence of a BcR engagement but with the help of a cytokine combination. This option is made likely by the results of some of the studies mentioned above, although more detailed tests using animal model systems are not available and the real efficacy of these agents in maintaining/expanding the CLL clone independently of BcR stimulation remains to be determined.

Certain mechanisms have been investigated in greater detail. To study the function of TLRs, CLL cells have been exposed to CpG in vitro, which engages TLR-9 on CLL cells. This causes some cell proliferation, accompanied by the apoptosis of variable proportions of cells [202,203]. Cytokines, particularly IL15, modulate the outcome of the stimulation, causing both the prolonged survival and proliferation of both U and M-CLL cells [202,204]. Based on these in vitro observations, it is possible that the activation of CLL cells, induced by microbial or autologous DNA, is followed by cell proliferation in the presence of adequate IL15 concentrations in vivo, a situation that can occur at sites where CLL cells are in contact with CD4^+^ T cells producing several cytokines, including IL15 [203]. This mechanism is apparently independent of BcR signaling since CLL cell proliferation is observed in vitro in the absence of any BcR cross-linking, although the role of the presence of self-antigens and of autologous stimulation is yet to be investigated. Thus, with the due caution, we may state that stimulation via native immunity receptors may also be part of the mechanisms promoting cell expansion in the presence of TKIs.

Co-culturing of autologous T cells, activated by exposure to anti-CD3 and anti-CD28 mAbs, induces a substantial proliferation of CLL cells. This phenomenon is blocked by the presence of anti-CD40 or anti-CD40L mAbs, indicating the requirements for these molecular interactions. However, exposure of purified CLL cells to fibroblasts transfected with CD40L does not have the same effect as activated T cells; CLL cells exhibit better in vitro survival but do not proliferate, suggesting the need for cytokines in order for proliferation to occur [193,205,206]. IL21 has been found to be a likely candidate for this function, given that T cells producing IL21 support CLL cell proliferation both in vivo and in vitro [207]. Notably, purified CLL cells pre-activated in vitro via contact with CD40L-bearing cells may undergo apoptosis rather than proliferation when exposed to IL21, indicating a complex interplay between cytokines and accessory cells in regulating clonal expansion [208]. Since the response to IL21 can be induced by the sole interaction with activated T cells, in the absence of BcR cross-linking, the growth of CLL cells can be considered BcR-independent and may also be TKI-independent [207].

The support provided by IL23 to CLL cell proliferation represents another example of a BcR-independent mechanism. U-CLL cells express low levels of the IL23R chain, which is non-functioning because of the absence of the complementary IL12Rβ1 chain required to form a fully functional IL23R complex. M-CLL cells do not express IL23R or IL12Rb1 chains. However, a substantial proportion of both U- and M-CLL cells can be induced to express an IL23R complex (formed by the two receptor chains) when co-cultured with activated autologous T cells or with CD40L-bearing fibroblasts [209]. In these conditions, CLL cells proliferate in response to IL23 in vitro and produce endogenous IL23. The production of IL23 activates an autocrine/paracrine loop, sustaining further cell proliferation. The expression of a fully functional IL23R complex and IL23 secretion by CLL cells is not observed following sIgM/sIgD cross-linking and is not affected by the blockage of BcR signaling by ibrutinib following stimulation with CD40-bearing fibroblasts, indicating BcR independence. Therefore, this may represent a valid mechanism for CLL cells to overcome the growth limitations imposed by TKIs in vivo [209]. Notably, the activation of the IL23/IL23R complex loop seems relevant for CLL clonal expansion in vivo, since inhibition of the loop prevents the taking-up of CLL cell grafts and causes the shrinkage of engrafted tumors in immunodeficient mice [209]. Thus, targeting the IL23 pathway with specific monoclonal antibodies or with miR-146b mimics, which are capable of downregulating the expression of the IL23R complex [210], may be explored as potential tools to overcome TKI resistance.

## 5. Disease Progression and Clonal Evolution

Many CLL cases are characterized by a progression to more advanced stages. Schematically, this progression involves different phases, including the transition from clinical MBL to full-blown CLL, followed by a phase of progressive bone marrow and lymphoid tissue invasion by leukemic cells, as can be assessed using the Rai or Binet staging systems, via an accelerated turn-over of leukemic cells and the increasing expression of cell activation markers [4,5,10]. In a minority of CLL patients (approximately 1% per year), this progression culminates in a deadly condition known as Richter transformation (RT), characterized by the appearance of diffuse large B cell lymphoma (DLBCL) or, more rarely, of Hodgkin lymphoma (HL) in a patient with CLL [211,212]. In most patients developing DLBCL (approximately 70%), the IGHV-IGHD-IGHJ gene rearrangement is identical in the DLBCL and in the CLL cells, implying a common cellular origin of the two lymphoproliferative disorders. In the remaining cases, the two rearrangements are different, indicating that the patient developed a de novo lymphoproliferation, apparently unrelated to CLL, although, as discussed below, alternative interpretations can be offered. The latter patients generally have a better prognosis and outcome than those with concordant rearrangements. Furthermore, in the patients developing HL, the Hodgkin–Reed–Sternberg (HRS) cells present rearrangements, which may be concordant (in a minority of patients) or discordant with that of the leukemic clone. The paucity of cases and the complexities of rearrangement determination in HRS cells have limited the analyses in this area and precise percentage values cannot be given [213]. Notably, the patients developing HL have better courses of disease and outcomes than those developing DLBCL.

BcR plays a relevant role in the clonal evolution underlying the above phenomena. For example, U-CLL presents a more rapid progression than M-CLL and there are differences in the progression of the groups of cases utilizing different stereotyped receptors, as already discussed. Moreover, RT is more frequent in cases with unmutated and/or stereotyped BcR. Indeed, the presence of these BcR features, together with *NOTCH1, TP53* and *CDKN2A* mutations or disruption and *c-myc* activation, indicates a propensity towards RT in patients with CLL, and the concomitant presence of these markers in a patient with RT is indicative of a dire outcome [214,215,216,217]. Notably, RT patients with concordant IGHV-IGHD-IGHJ gene rearrangements between CLL and DLBCL are more likely to have unmutated/stereotyped rearrangements [218], in particular belonging to subset #8, whereas DLBCL with discordant IG gene rearrangements can originate in a larger proportion of M-CLL patients. RT characterized by the onset of HL can originate in patients with both M- and U-CLL, although the paucity of cases makes it difficult to determine percentage values [213]. Finally, the response, albeit frequently a partial response, to the Bruton TKIs ibrutinib and acalabrutinib reported in a few patients with RT suggests that the cell-proliferation-promoting function exerted by BcR also continued following RT [219,220,221].

Clonal evolution can be characterized by structural changes aimed at adapting the BcR to a more efficient promotion of cell proliferation. To clarify this issue, the onset of CLL subclones with changes in the BcR combining site has been investigated. Initially, IGHV-IGHD-IGHJ gene cloning and Sanger sequencing methodologies were employed to detect the appearance of new mutations. Despite the known limitations of this approach, researchers were able to detect subclonal variants of the original rearrangements and the mutation pattern provided evidence for antigenic stimulation/selection in some instances [222,223,224]. A particularly pronounced accumulation of mutations in both the IGHV-IGHD-IGHJ and IGLV-IGLJ gene rearrangements was reported in CLL cases utilizing stereotyped subset #4 and expressing an IgG isotype [225]. The advent of NGS methodologies allowed researchers to achieve considerable sequencing depth and to minimize amplification biases in the identification of subclones with the same IGHV-IGHD-IGHJ gene rearrangements as the original leukemic clone [226,227]. This approach demonstrated intraclonal diversification in virtually all CLL clones investigated, irrespective of whether they were initially classified as U or M. In some cases, the subclonal variants were substantial, reaching the numerical criteria of MBL. Analyses of the mutation pattern excluded the possibility that they were related to a process of antigen stimulation/selection. Furthermore, these mutations did not appear to be mediated by activation-induced deaminase (AID) and by error prone Pol*η*, which operate in germinal center B cells to ensure antibody affinity maturation [228], suggesting that a fraction of the leukemic clone could become less dependent on the recognition of specific epitopes by the BcR. Therefore, in addition to mutations that improve the binding of BcR to its ligands, other mutations could facilitate stimulation via BcR through alternative signaling mechanisms, including autonomous signaling or stimulation by superantigens binding to BcR framework structures. However, it remains to be ascertained whether the numerous clonal variants detected via NGS are progressively lost or whether they contribute to clonal evolution, causing an enhanced disease progression (and if so, to what extent do they contribute). These considerations are in line with the data obtained with whole-genome or whole-exome sequencing, showing that CLL has a low load of mutations and that the accumulation of these mutations is mostly unrelated to the enzyme activities which operate in the GC to favor antibody affinity maturation and isotype switching [24]. In connection with this, it is of interest that AID has been found only in a small proportion of CLL cells which reside in the fractions of recently activated cells in the proliferating centers. However, the issue of whether this AID activity participates in clonal evolution, causing disease progression, or represents a transient activation marker is an unresolved problem [229].

Based on the above considerations, it is likely that the enzyme machinery leading to antibody affinity maturation is not involved in the origin of other lesions emerging with disease progression, such as *NOTCH1, BIRC3, SF3B1* and *TP53* mutation/disruption, and thus does not represent a relevant clonal evolution event. Notably, the availability of NGS methodologies has led to the demonstration that some of these alterations can be present at the subclonal level in a substantial fraction of early CLL cases [23,230,231,232,233]. Moreover, the appearance of these mutations may be facilitated by events promoting cell proliferation such as those exerted by BcR, given that most of these cytogenetic lesions are more frequently observed in U-CLL cases and in CLL clones utilizing stereotyped BcRs.

## 6. The Cell of Origin of CLL

If we assume that leukemogenesis, like other processes of malignant transformation, is a multistep phenomenon, with a progressive accumulation of transforming mutations, the next question is where does this process begin in B cell ontogeny? It may begin in mature B cells, since their capacity of proliferating in response to external stimuli makes them a suitable target for subsequent transforming events. Alternatively, the initial transforming mutation(s) can occur in the hematopoietic stem cells (HSCs) before differentiation along the B cell line is initiated. The presence of these mutations is likely to influence the process of B cell maturation and to facilitate the transforming events. The two models are dealt with separately here.

### 6.1. Origin from Mature B Cells

This process is thought to begin with a cell expressing a fully functional BcR, the engagement of which may facilitate the accumulation of transforming mutations. Murine models focused on B-1 B cells, which generate lymphoproliferative disorders similar to human CLL, support this mechanism. Using cell transfer techniques and allotype markers to identify the cell source of the IG, it has been shown that B-1 B cells, transferred from adult to neonatal mice, expand in the host, leading to oligoclonal or monoclonal B cell accumulations, which are thought to be the immediate progenitors of leukemic cells. The generation of a full-blown leukemic process was accelerated when the transfer was carried out using mice that were transgenic for *tcl1* [151,234,235]. Notably, in most of these murine models, BcR specificity turned out to be a key element in leukemogenesis when the issue was investigated further. For example, in the *tcl1* transgenic model, the BcR of the leukemic cells has a structural resemblance and a similar specificity to that of human U-CLL, including the use of unmutated IGHV genes, stereotypy, poly-specificity and also specificity for self-antigens such as non-muscle myosin II A (MyIIA) [151,236,237]. In addition, genetic or drug-induced inactivation of BcR signaling in *tcl1* mice causes a substantial delay in the onset of leukemia [238], whereas a more efficient BcR signaling, obtained via genetic inactivation of the PTEN molecule, which has a negative regulatory effect on BcR signaling, results in an accelerated onset of leukemia [239]. Notably, *pten* gene deletion is also observed in CLL, further indicating that more efficient BcR signaling promotes clonal expansion in patients as well [239]. Finally, the insertion of genes encoding for a fully functional BcR with anti-MYIIA or anti-phosphatidylcholine specificity into mice resulted in the onset of a CLL-like disease which was greatly facilitated by a *tcl1* background [103,236]. The leukemia developed in these mice presented with cytogenetic lesions reminiscent of the human del13q seen in CLL patients [236]. Thus, an active collaboration between BcR engagement and transforming gene lesions seems to be required for leukemogenesis in these murine models. In order to transfer the notions derived from murine models into the study of human CLL, certain conditions must be fulfilled, namely, the identification of a homogeneous population of B-1 B cells and/or of a cellular subset with an IG gene repertoire resembling that of CLL.

As already discussed, a human B cell population sharing a BcR repertoire with that of murine B-1 B cells and CLL cells has not been found. This holds true for normal CD5^+^ B cells as well, which have been considered the most likely CLL progenitor cells for a long time. In the absence of a demonstration of such a B cell subpopulation, we must assume that the process of leukemogenesis is characterized by repertoire selection, which takes place through BcR engagement occurring concomitantly with an accumulation of transforming mutations, and that the process may begin in any B cell equipped with a functional BcR. However, before proposing models for the origin of CLL from mature B cells, we must discuss whether U- and M-CLL are generated from the same or different cells. Initial studies with gene and miRNA expression profile analyses demonstrated similarities between the cells from the two groups, suggesting a common cellular origin [240,241,242]. On the other hand, later observations supported a different cellular origin, because of their repertoire differences and their utilization of different sets of stereotypes in the U- and M-CLL groups. Moreover, CLS-IG from normal B cells share the characteristic IGHV mutational status observed in the corresponding CLL stereotypes (i.e., CLL stereotypes mutated in CLL also are mutated in the CLS-IG of normal B cells, and the same has been observed for the unmutated stereotypes) [72]. Finally, normal B cells present different methylation patterns which correlate with their maturation stage. U- and M-CLL cells present methylation patterns consistent with origins from naïve or memory B cells, respectively, which may be indicative of differences in the cells in which leukemogenesis started [132,243]. Altogether, these observations favor the hypothesis that the malignant transformations of U- and M-CLL cells follow different trajectories. Based on these considerations, it is plausible that U-CLL originates from unmutated virgin B cells, which could be follicular mantle B cells, B cells at the T cell/B cell zone border or unmutated marginal zone (MZ) B cells located in the spleen or in the corresponding anatomical areas of different lymphoid tissues [244]. The transformation process may begin in transitional B cells, which are the immediate progenitor of naïve B cells [245]. M-CLL is likely to derive from post-GC B cells or mutated MZ B cells (see Figure 4A).

In principle, the process of the transformation of mutated M-CLL cells could commence with a naïve B cell, as suggested in [93]. However, for this hypothesis to be tenable, we have to assume that during transformation a complex series of events leading to maturation into memory cells occurs, a process which is hard to imagine, particularly as an antigen-driven process, as is probably required in this specific case.

Irrespective of where and when the transforming events occur, it is conceivable that some of them contribute to a facilitation of cell stimulation via BcR.

### 6.2. Origin from HSCs

This model implies that the initial transforming event(s) occurs in HSCs, which are capable of originating all blood lineages. Because of the transformation, the HSC gives rise to clonal hematopoiesis, characterized by a predominant differentiation towards the B cell lineage. This event leads to the generation of B cell subclones with different IGHV-IGHJ-IGHD rearrangements, each of which is similar to MBL. One of the MBL cells subsequently transforms into a full-blown CLL cell [246,247]. Support for this hypothesis was initially provided by the observation of an abnormal expansion of pro-B cells in the bone marrow of CLL patients. Subsequently, it was shown that purified HSCs from M-CLL patients xenografted in immunodeficient mice could generate accumulations of polyclonal pro-B cells and of oligoclonal mature CD5+ B cells with features of MBL, as assessed via the determination of their IG gene rearrangements [248]. The rearrangements of B cells in xenografted mice were different from that of the CLL clone of the HSC donor, suggesting that the process of maturation of xenografted “transformed” HSCs into B cells was different, owing to the different microenvironment [248] (mouse vs human) in which the selection via BcR engagement was taking place. Two additional findings are also of interest: (i) in the experimental conditions used, the CLL clone of the patient, the donor of the HSC, was not expanded, probably because the experimental protocol excluded the injection of autologous T cells into mice, which are required for the engraftment of mature CLL cells [249]. Therefore, the confounding effect possibly exerted by the presence of the patients’ mature CLL cells could be eliminated. (ii) Cytogenetic lesions like those of the patients’ CLL clone (e.g., TP53 mutations, del 17p- or del 13q-) were not observed in the engrafted B cells. This, on the one hand, suggests that the B cells expanding in mice did not yet develop those cytogenetic lesions typical of full-blown CLL. On the other hand, the absence of shared cytogenetic lesions made it difficult to ascertain whether the B cell clones that originated in mice were descendants of or related to the same clone giving rise to CLL.

This model has several merits, as it draws parallels between CLL and myeloproliferative disorders of the elderly, such as polycythemia vera and thrombocythemia, which are characterized by clonal hematopoiesis with a predominant differentiation along a single cell lineage [250], and underlines the relevance of the mutations in HSC with advancing age [251]. The model contributes to the explanation of cytopenias affecting lineages other than B cells in progressing CLL, as, at this stage, most hematopoiesis could be clonal, i.e., carried out by “transformed” HSCs. The notion of clonal hematopoiesis is also compatible with the frequent observation of oligoclonal B cell accumulations, in addition to the major B cell clone, in both MBL and CLL [252,253,254] and explains the relatively frequent occurrence of DLBCL and HL with IGHV-IGHD-IGHJ gene rearrangements discordant from that of CLL cells. These lymphoproliferative disorders would, in fact, originate from the same HSC, giving rise to the MBL/CLL cell clone, and their emergence could be favored by a defective tolerance mechanism during B cell maturation, facilitating stimulation by self-antigens. Analogous mechanisms could be invoked to explain the particular features of the BcR repertoire in CLL and failures in tolerance induction would account for the high frequency of auto-immune phenomena, including the BcR-anti-BcR reactions at the basis of autonomous signaling. Finally, the notion that CLL is derived from an initial “transformed” HSC would perhaps provide a wider range of cellular/molecular targets to test our understanding of the origin of familial CLL, in which several gene lesions seem to be implicated [33,34,255,256].

Problems with this model are generated by the fact that molecular lesions, shared by the MBL/CLL clone and normal lymphocytes, or blood cells other than lymphocytes, are difficult to demonstrate consistently in a large number of patients, as the available studies have shown and discussed [257,258,259,260]. The same difficulties exist regarding the demonstration of a shared cellular origin between patient CLL cells and the oligoclonal B cell accumulations that originated in mice xenografted with HSCs from the same CLL patient.

## 7. Conclusions

The information on the IGHV and IGLV gene repertoire of CLL clones has had a noticeable impact on the knowledge of CLL. Repertoire selection may facilitate the onset of leukemia by providing the optimal promoting effect for B cell expansion and the propagation of transforming mutations to the cellular progeny. This is particularly true if transformation includes events that facilitate stimulation via BcR. The need for this selection process may help to explain the differences between the normal and leukemic BcR repertoires. Several modalities of BcR engagement have indicated that CLL clones are continuously exposed to a variety of signals that enable their survival and expansion. The knowledge of BcR repertoires has also contributed to elucidating further the differences between the two major CLL groups, i.e., U- and M-CLL, and to explaining in part their different clinical behaviors and different trajectories of leukemogenesis. The knowledge of the pathogenetic role of the BcR-mediated signaling has made it possible to find suitable targets of biological therapies capable of inhibiting clonal expansion. These represent a major tool to control the disease, although they are not the definitive therapeutic answer, since unresponsiveness to drugs can be initiated due to target mutations and/or to the utilization of stimulation pathways that are independent of BcR signaling. Finally, repertoire development can provide information on the ontogeny of CLL, although the data accumulated so far have not yet permitted researchers to ascertain which of the models proposed is preferable and which fine mechanisms are involved.

## Figures and Tables

**Figure 1 ijms-23-14249-f001:**
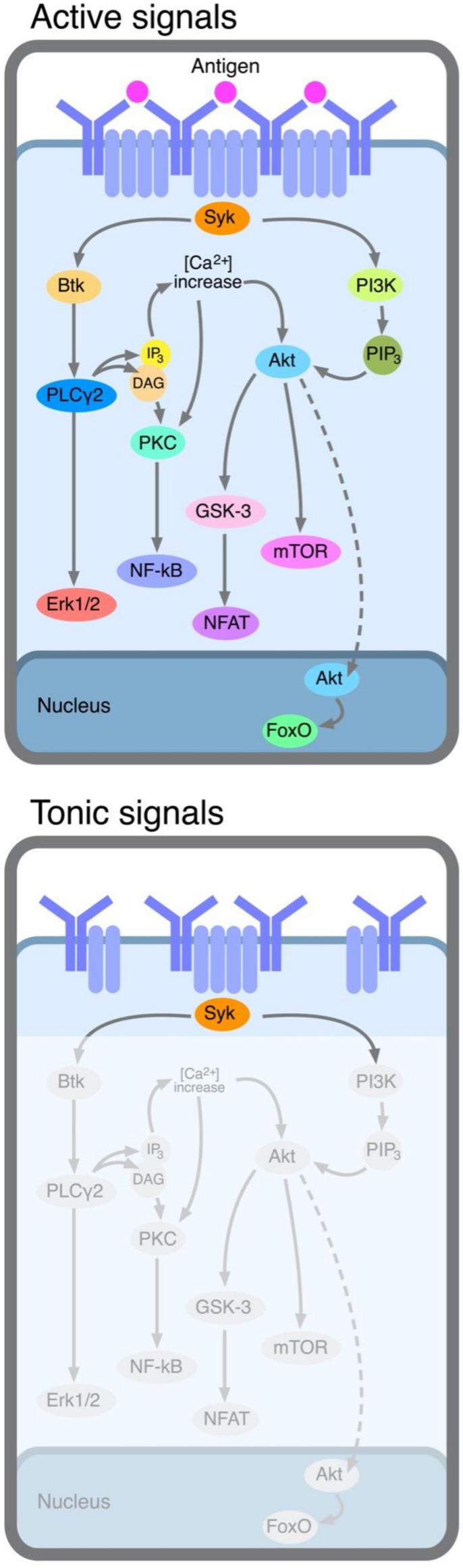
Active and tonic signals delivered by BcRs to CLL cells. *Active signals (top).* The BcR at the surface of CLL cells is cross-linked together with its accessory molecules (CD79a and CD79b, indicated in pale blue) by a self-antigen or an antigen of microorganisms, causing the activation of the downstream signaling pathway, the major steps of which are schematically depicted. *Tonic signals (bottom).* Spontaneous micro-aggregation of the BcR and its accessory molecules leads to Syk activation, which in turn activates downstream signaling. This further step is represented in gray, given that the steps involved have not yet received experimental confirmation.

**Figure 2 ijms-23-14249-f002:**
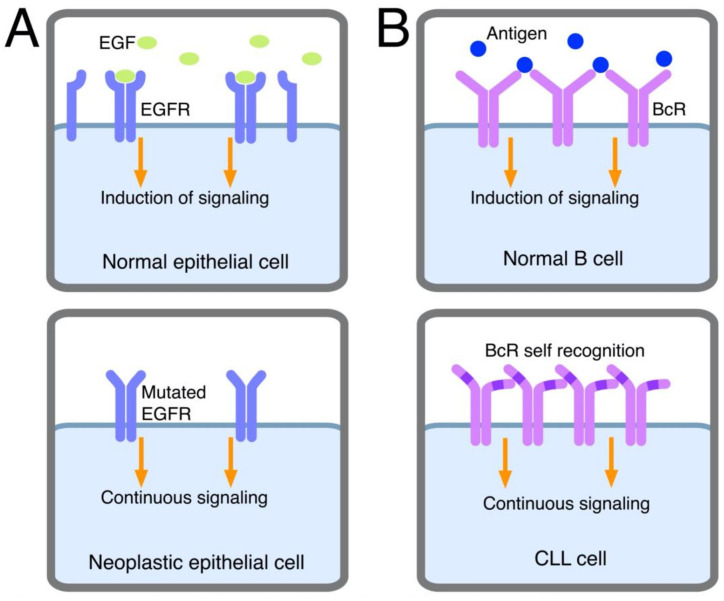
Autonomous signaling in CLL cells. The mechanisms leading to autonomous signaling ((**B**), right) are depicted and compared with the continuous signaling delivered by mutated EGFR in cancer epithelial cells ((**A**), left). In normal epithelial cells, dimerization of a proportion of the membrane’s EGFR molecules by EGF causes the activation of the downstream pathway, which is transient and lasts until EGF is present in the micro-environment (top). In neoplastic cells, “mutated” EGFR molecules aggregate spontaneously, providing continuous signaling to the cells which are induced to proliferate (bottom). In normal B cells, BcR stimulation is caused by the presence of a specific antigen and is transient (top). In CLL, in which autonomous signaling occurs, BcR signaling is constantly activated since the BcR binds to a specific structure of the BcR itself (marked in blue).

**Figure 3 ijms-23-14249-f003:**
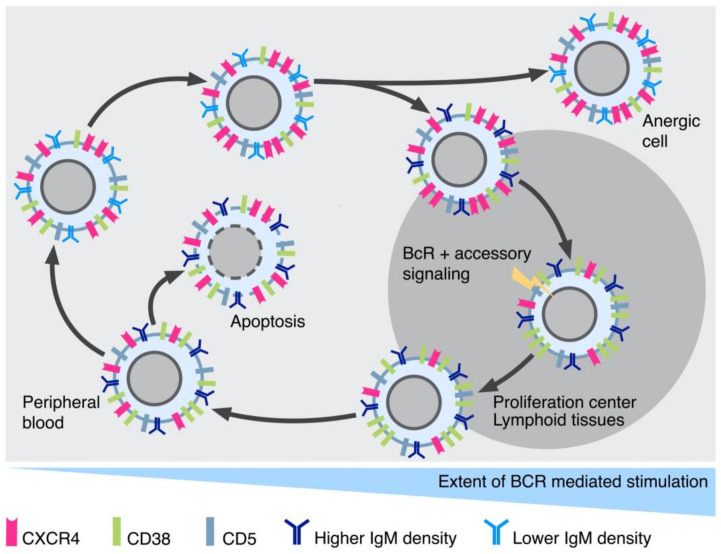
Hypothetical model of CLL cell turn-over. CLL cells are stimulated via BcR through the mechanisms depicted in Figure 1 and Figure 2. The cells proliferate with the assistance of accessory cells and cytokines that are abundant in proliferating centers. Cell activation consequent to stimulation causes the downregulation of adhesion molecules, particularly of CXCR4, so that the cells can leave the proliferating centers and reach the circulation. Here, there is a progressive downregulation of adhesion molecules and upregulation of CD5. Concomitantly, activation molecules are downregulated, confirming the progressive tendency of CLL cells to reach a more resting state (here only CD38 is depicted among the activation molecules). A number of CLL cells undergo apoptosis outside the proliferating centers in the absence of accessory signals (this is particularly true for U-CLL cells). The CLL cells, progressively more quiescent because of the absence of stimulation, may become tolerant (particularly M-CLL cells) and/or upregulate a new set of adhesion molecules which allow re-entry into lymphoid tissues and initiate a new proliferation cycle in proliferating centers. The sIgM density varies across the different stages.

**Figure 4 ijms-23-14249-f004:**
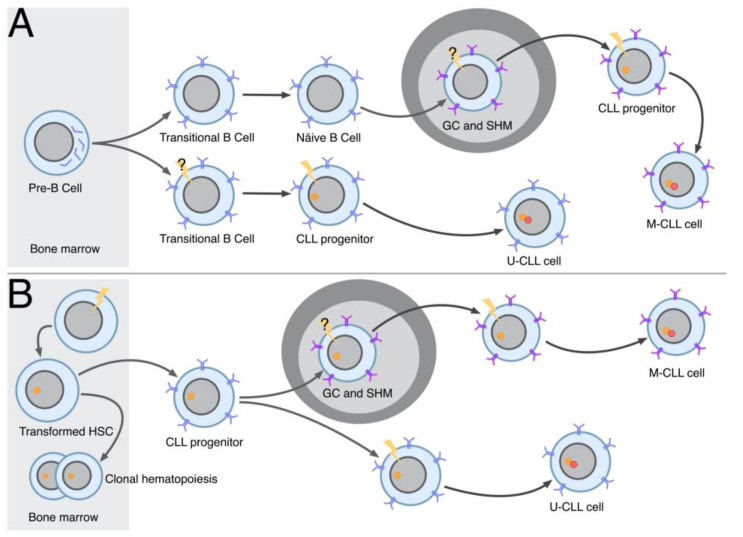
Models of CLL leukemogenesis. (**A**) *Origin of CLL from B cells expressing a fully assembled BcR.* The pathways of the origins of M-CLL (top) and U-CLL (bottom) cells are portrayed separately. The multistep transformation of M-CLL may be initiated already in the germinal center (GC), where somatic hypermutation (SHM) occurs (this step is indicated with a question mark). The transformation of U-CLL cells may be initiated already in transitional B cells (again indicated with a question mark), although it can certainly take place in naïve cells, which become progenitor CLL cells. Most of the transformation/selection process is promoted by BcR engagement. (**B**) *Origin of CLL from hematopoietic stem cells.* An HSC becomes a transformed HSC in the bone marrow and starts to generate oligoclonal CLL cell progenitors with features of mature B lymphocytes, which are subsequently exposed to additional transforming events. Among these transformed cells, one is eventually selected and this leads to full-blown CLL. This process of further transformation/selection promoted by BcR engagement occurs outside the GC for U-CLL cells, whereas it includes a passage in the GC for M-CLL cells. As discussed in the text, transformed HSCs are responsible for clonal hematopoiesis, predominantly giving rise to B cells but also including other lineages.

**Table 1 ijms-23-14249-t001:** General requirements for two CLL BcRs to be considered to belong to the same stereotyped subset.

Feature	Requirement
HCDR3 aa Identity	≥50%
HCDR3 aa similarity	≥70%
HCDR3 aa composition *	shared subset specific motifs
HCDR3 aa length *	shared
IGHV	shared clan or, in some cases, gene
IGHV mutational status	shared

* Satellite subsets have been recently introduced [56]. These are defined as IG gene rearrangements sharing the same HCDR3 motif with a different HCDR3 length and/or showing an offset of the HCDR3 motif. Thus, satellite subsets remain immunogenetically related to a main subset.

## Data Availability

Not applicable.

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
