# Peer review of "Old and New Facts and Speculations on the Role of the B Cell Receptor in the Origin of Chronic Lymphocytic Leukemia"

_ijms, 2022, doi:10.3390/ijms232214249_

Round 1
Reviewer 1 Report
The focal point of this revie was compiling available research and speculation towards identifying the Ig gene repertoire in the event of CLL and how does the B-cell receptor(BCR) play a role in maintaining and expanding the clonal population. The authors delve deeper into understanding what is currently known about the specification and features of BCR repertoire in case of CLL vs a normal B cell population. CLL cells are observed to exhibit certain subset of stereotyped combination/gene arrangements of V-D-J and V-J which in turn is speculated to be naturally selected to promote the rate of clonal expansion.
However, the authors were thorough to present the possible caveats currently present in the field that accompanies when comparing BCR on CLL and normal circulating B cells considering the mixed nature of population cohort often studied. The authors then move on to delve into compiling the story on what is known with BCR-mediated CLL cell stimulation through tonic (primarily for survival signal) and active signaling (proliferation and clonal expansion). The authors cover how ligand-independent (“tonic”) and ligand-dependent BCR signaling have been characterized, which can involve mutations of BCR pathway components or be triggered by (auto-) antigens present in the tissue microenvironment. Finally, the authors close the compilation with presenting known evidence on the aftermath of the BCR signalizing in these CLL cells.
The revie is an extensive compilation of the known research in understanding how BCR plays a role in CLL based on structural restrictions of the BCR, and BCR-dependent survival and growth of the malignant B cells.
The authors have done a beautiful job in extensively putting together with what is known in the field with proven facts and speculations carried out with BCR-mediated regulation of CLL.
Author Response
The reviewer did not ask for changes. However, we would like to thank him/her for the very lucid summary of the article and for the positive comments.
Reviewer 2 Report
Overall, this is a high-quality manuscript that has implications for the study of the engagement of the B cell Receptor (BcR) on the surface of leukemic cells which represents a key event in chronic lymphocytic Leukemia (CLL) leukemogenesis. The authors make a systematic contribution to the research literature in this area of investigation. There are only two issues associated with the Publication of the paper with the specific comments follow.
1. Bruton’s tyrosine kinase (BTK) and phosphoinositide 3-kinase (PI3K) are parallel downstream in the B-cell receptor (BCR) signaling pathways which are considered potential therapeutic targets for the treatment of B-cell lymphomas including CLL. These data emphasize the importance of PI3K as a therapeutic target in B cell lymphoma and additionally confirm that the combination with a BTK inhibition acts synergistically disrupting the BCR pathway. Therefore, a combination of targeting of the two pathways shows a promising therapeutic approach in the treatment of CLL.
2. The BcR repertoire of CLL may be different from that of normal B cells because of the presence of stereotyped sequences and of preferential usage of certain gene segments. Please discuss how the somatic IGHV hypermutation (SHM) and isotype switch take place in the BcR repertoire and by correlations existing between certain BcR features and the clinical outcome of CLL patients in the disease progression to DLBCL.
Author Response
We would like to thank him/her for the positive evaluation.
Comment 1. We have added a sentence commenting the function of different TK in the BcR signaling and the possibility that some of these TK can be in parallel pathways, giving the possibility of a combined use of inhibitors, if compatible with the adverse effects.
Comment 2. Indeed, the issue of clonal evolution and of its relevance in all aspects of disease progression, including RT, was not emphasized in the preceding version of manuscript. Therefore, we have added a new section (Disease progression and Clonal evolution) to deal specifically with this subject. In addition, part of the following section on the existing hypotheses on the cell of origin of CLL has been modified to provide a further possible explanation for the origin of a second lymphoproliferative disorder, like DLBCL or HL, in CLL patients. We believe that the addition of these parts has improved the information conveyed by the paper and we thank the reviewer for the suggestion.